# $Mn^{2+}$ induced significant improvement and robust stability of radioluminescence in $Cs_3Cu_2I_5$ for high-performance nuclear battery

Xiaoming Li[1], Jiaxin Chen[1], Dandan Yang[1], Xi Chen [1], Dongling Geng[1], Lianfu Jiang[1], Ye Wu[1], Cuifang Meng[1] & Haibo Zeng [1 ✉]

Fluorescent type nuclear battery consisting of scintillator and photovoltaic device enables semipermanent power source for devices working under harsh circumstances without instant energy supply. In spite of the progress of device structure design, the development of scintillators is far behind. Here, a $Cs_3Cu_2I_5$: Mn scintillator showing a high light yield of ~67000 ph $MeV^{-1}$ at 564 nm is presented. Doping and intrinsic features endow $Cs_3Cu_2I_5$: Mn with robust thermal stability and irradiation hardness that 71% or >95% of the initial radioluminescence intensity can be maintained in an ultra-broad temperature range of 77 K-433 K or after a total irradiation dose of 2590 Gy, respectively. These superiorities allow the fabrication of efficient and stable nuclear batteries, which show an output improvement of 237% respect to the photovoltaic device without scintillator. Luminescence mechanisms including self-trapped exciton, energy transfer, and impact excitation are proposed for the anomalous dramatic radioluminescence improvement. This work will open a window for the fields of nuclear battery and radiography.

---

[1] MIIT Key Laboratory of Advanced Display Material and Devices, School of Materials Science and Engineering, Nanjing University of Science and Technology, Nanjing, China. ✉email: zeng.haibo@njust.edu.cn

n some particular applications, such as equipment for outer planet exploration and undersea detection, energy supply is usually challenging due to the absence of light and difficulties in charging or replacing power sources. These facilities may operate continuously for more than 10 years at places without human beings and this calls for energy systems that can generate energy themselves and work for a long time. Nuclear batteries (NBs)[1], which convert energy from radiation sources into electricity appeal to above applications due to their extremely long working lifetime and high stability against harsh conditions. Recently, the radioluminescent NBs have been researched extensively due to the possibilities of higher energy conversion efficiency and protection of terminal semiconductors[2,3]. Such kind of battery consists of three parts, namely, radiation source, scintillation layer, and photovoltaic device (PVD). The scintillation layer converts radiation energy into light emission and then, the PVD converts these photons into electricity. There is no doubt that the high efficiency of scintillator, radioluminescence (RL) spectrum consistent with the response of PVDs, perfect coupling between luminescent materials and PVDs are the dominating factors to achieve high conversion efficiency. Besides, a good stability of the scintillator against radiation and environmental attack is the basis of NB's stability.

In the past decade, investigations on the design of the device structure[4–6], application of various scintillators (organic or inorganic ones)[7], effect and optimization of physical parameters[8–10], and coupling between scintillators and PVDs have been carried out to improve the NB efficiency[11]. Despite the power output and efficiency of NBs have been improved significantly, the scintillators employed in these researches were mainly conventional transition or rare earth metal doped phosphors[7,12,13], which typically possess short emission wavelength or low light yield (LY). For standard PVDs, the conversion efficiency at short wavelength (mainly shorter than 450 nm) region is usually low. Though some scintillators can emit long-wavelength light, their LYs are commonly frustrating. Therefore, to further improve the performance of NBs, new scintillators with both long wavelength and high LY are urgently needed to break the bottleneck.

The success of lead halide perovskites in light-emitting applications redraw our attention back to metal halide scintillators[14–16]. Their high photoluminescence quantum yield (PL QY) and good radiation absorption ability may endow them with high LY. Upon the first report of RL phenomenon of perovskite nanocrystals (NCs)[17], demonstration of X-ray imaging based on perovskite NCs has been investigated extensively[18–20]. Though the emission wavelength of perovskite NCs can be modulated feasibly, their LY is indeed low compared with many commercial scintillators[21]. Besides, their poor stability and toxicity issues make it challenging in practical applications. Referring to the problem of lead, Tang et al. developed a new copper-based scintillator with ultrahigh LY[22,23]. Whereas, the emission peak locates in the UV region, which is not consistent with the response of PVDs. Kovalenko et al. reported the fabrication of zero-dimensional tin (Sn$^{2+}$) halides and achieved relatively high LY and long-wavelength emission (660 nm) simultaneously[24]. But, Sn$^{2+}$ is easily oxidized in air and the organic part may also be destroyed at high temperature. Very recently, a new kind of zero-dimensional copper halide (Cs$_3$Cu$_2$I$_5$) NC was reported to exhibit high QY and good stability with self-trapped exciton (STE) based blue emission[25–30], making it potential candidate of highly efficient and stable scintillator for NBs[31].

Here we show that doping Mn$^{2+}$ into Cs$_3$Cu$_2$I$_5$ crystals can induce significantly improved RL efficiency compared to intrinsic Cs$_3$Cu$_2$I$_5$ crystals and a maximum LY of ~67,000 ph MeV$^{-1}$ at an emission wavelength of 564 nm was achieved. This is, to the best of our knowledge, almost the highest value at such a long

emission wavelength based on low cost material[32]. Furthermore, the Mn doped Cs$_3$Cu$_2$I$_5$ (Cs$_3$Cu$_2$I$_5$:Mn) exhibits good stability against temperature variation that 71% of initial RL intensity can be maintained in an ultra-broad temperature range of 77–433 K while the RL intensity of pure Cs$_3$Cu$_2$I$_5$ shows a severe deterioration of 92.5%. Additionally, when Cs$_3$Cu$_2$I$_5$:Mn was irradiated continuously for one month, more than 95% of its initial RL intensity has remained after a total dose of ~2590 Gy. To figure the reasons for the abovementioned phenomena, the luminescence mechanism is discussed detailedly with thorough analyses of structural and luminescent properties. These superiorities imply the promising application of the fashionable copper halide in NBs. Efficient and stable NB based on Cs$_3$Cu$_2$I$_5$:Mn and PVD was fabricated finally, which shows an output improvement of 237% respect to that without scintillator and a power output of 0.46 μW cm$^{-2}$. It is expected that such a new and robust scintillator will open a window for the fields of NBs and radiography.

## Results

**Structure characterization of Mn$^{2+}$ doping.** Cs$_3$Cu$_2$I$_5$ consists of zero-dimensional electronic structures, and Cu$^+$ ions occupy two sites, a tetrahedral (Cu 1) and a trigonal (Cu 2) site (Fig. 1b). According to previous theoretical calculations, Cu 3d orbitals dominantly contribute to the valence band maximum (VBM), while the conduction band minimum (CBM) is mainly composed of Cu 4s and I 5p orbitals[25]. Therefore, the exciton transition is localized, which mainly takes place within the Cu–I structures. Then, to modulate the electronic structure and optical properties, replacing Cu$^+$ or I$^-$ ions becomes the major strategy. In this work, we aim to introduce an additional exciton recombination center in this new copper halide by doping Mn$^{2+}$ to prolong the RL wavelength or even improve the RL efficiency. Though the possibility of achieving long emission wavelength has been evidenced very recently by doping Mn$^{2+}$ ions[28,29], the scintillation performance and detailed structural/optical analyses have never been reported.

Theoretical calculations were firstly conducted to evaluate the influences of Mn doping. We can see that replacing either the tetrahedral or the trigonal Cu ions results in similar electronic structures and system energy with a difference of ~90 meV even at a heavy doping concentration (Supplementary Fig. 1). Mn dopant introduced additional energy levels between the band gap with a difference of ~94.7 meV for different doping sites. Moreover, the comparable ion size of Cu$^+$ (0.6 Å) and Mn$^{2+}$ (0.66 Å) makes the substitution process to happen easily. Therefore, replacement of Cu ions with Mn ions is possible from both crystal and energy factors.

Cs$_3$Cu$_2$I$_5$:Mn was prepared according to a previous work with significant modifications[26]. Supplementary Figure 2 shows the X-ray diffraction (XRD) profiles for the Cs$_3$Cu$_2$I$_5$:Mn with different doping concentrations, which exhibit similar diffraction peaks well indexed by standard data. For the sake of simplicity, Cs$_3$Cu$_2$I$_5$:Mnx% was named as Mnx% in the following discussions, where x is the nominal concentration. The diffraction peaks first shift to larger degree upon doping and then shift to smaller degrees with the increase of Mn$^{2+}$ concentration (Fig. 1a). The abnormal lattice contraction results from additional chloride ions in the precursor, which is much smaller than iodide ions. It should be noted that the incorporation of chloride ions does not change the electronic structure significantly (Supplementary Fig. 3). Besides, the concentration of chloride ion cannot be too high due to the intrinsic lattice instability[33]. The lattice expansion with increased Mn$^{2+}$ concentration can be assigned to the larger ion radius of Mn$^{2+}$ [34]. When doping concentration increases to 25%, the diffraction peak shifts back to a higher degree, indicating

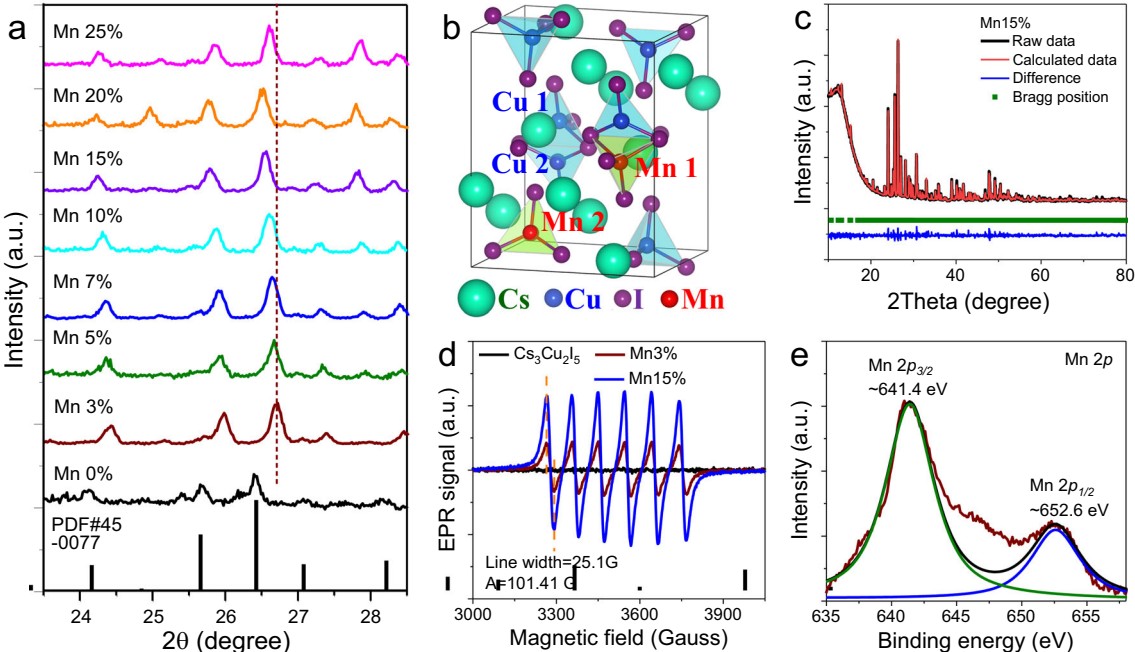

**Fig. 1 Structural analyses of Mn doping in $Cs_3Cu_2I_5$. a** Partial XRD patterns of powders with different doping concentrations. **b** Schematic demonstration of crystal structure of $Cs_3Cu_2I_5$ and doping sites. **c** Rietveld refinement of Mn15%. **d** EPR results of Mn3% and Mn15%. **e** XPS curve of Mn 2p in Mn15%.

the saturation of $Mn^{2+}$ and lattice instability at high doping concentration, which will lead to poor optical properties undoubtedly.

To verify the local crystal environments of $Mn^{2+}$ ions in the lattice, rietveld refinements were performed for Mn3% (Supplementary Fig. 4) and Mn15% (Fig. 1c), which confirmed the occupation of $Mn^{2+}$ ions at both $Cu^+$ sites (Supplementary Table 1). Electron paramagnetic resonance (EPR) and X-ray photoelectron spectroscopy (XPS) measurements were also conducted. No EPR signal is detected in the intrinsic sample while six well-resolved spectral lines corresponding to $^{55}Mn$ nucleus ($I = 5/2$) can be observed from both the Mn3% and Mn15% samples (Fig. 1d)[35]. First of all, such a strong and high resolved sextet spectral feature suggests a uniform dispersion of $Mn^{2+}$ in lattice sites with multi-coordination rather than on the surface. This also rules out the formation of other $Mn^{2+}$ related material phases[36]. Second, the g factor of 2.0053, hyperfine splitting constant ($A = 10.14$ mT), and the spectral linewidth (2.51 mT) keep the same in these two samples, indicating the stationary coordination state of $Mn^{2+}$ in the lattice despite the signal intensity variation resulted from increased concentration. The A in these samples is a bit larger than that in other metal halides, which might be derived from the low coordination number and large anion radius[37]. XPS results indicate the maintenance of $Cu^+$ state that a 2p doublet with the binding energy of 932.1 eV ($2p_{3/2}$) and 951.8 eV ($2p_{1/2}$) is observed (Supplementary Fig. 5)[22]. Similar $Cs^+$ and $I^-$ ion chemical states are obtained in both undoped and doped samples (Supplementary Fig. 6). Mn core level spectrum shows Mn $2p_{3/2}$ peak at 641.4 eV and Mn $2p_{1/2}$ peak at 652.6 eV with a separation of 11.2 eV, which confirm the existence of $Mn^{2+}$ dopant (Fig. 1e). The scanning electron microscopy (SEM) image and elemental mapping results (Supplementary Figs. 7 and 8) reveal the particle size of several to a dozen micrometers and Cs, Cu, I, Mn elements are homogeneously distributed.

**Optical and anomalous scintillation properties.** Intrinsic $Cs_3Cu_2I_5$ exhibits bright blue emission under a UV lamp (254 nm, Fig. 2a inset), which agrees well with the re-absorption free PL spectrum peaking at ~442 nm (Fig. 2a)[25]. Considering the

zero-dimensional electronic structure and related large exciton binding energy, the quantum size effect is negligible in this material and large crystals can usually achieve high QY easily due to low lattice defect density and re-absorption free characteristic[26,31]. The prepared $Cs_3Cu_2I_5$ powder exhibits a QY of 38.3%. Doping of $Mn^{2+}$ introduces another emission band centered at 556 nm along with the intrinsic blue emission (Fig. 2b). When measuring the PLE spectrum with a monitor wavelength of 442 nm (intrinsic emission), it exhibited a similar spectrum to that of undoped sample. Representative excitation spectra monitored around 556 nm are shown in the bottom part, Fig. 2b, which exhibit three bands centered at 310 (intrinsic), 385 ($^6A_1(^6S)\rightarrow^4T_2(^4D)$), and 485 nm ($^6A_1(^6S)\rightarrow^4T_1(^4G)$), respectively. The UV part and that monitored at intrinsic emission wavelength are semblable, indicating the yellow emission from $^4T_1(^4G)$ to $^6A_1(^6S)$ can be achieved by energy transfer (ET) from $Cs_3Cu_2I_5$ or direct excitation of $Mn^{2+}$. Additionally, the similar shape of PLE spectra with different monitor wavelength indicates the single excited state. The inset in Fig. 2b shows the image of the sample under a UV lamp (365 nm) with dazzling yellow light. Figure S9 shows the PL spectra with different doping concentrations. The yellow emission intensity increases with the increase of doping concentration and reaches a maximum at Mn15%. Then, the intensity begins to decline when further elevating the doping concentration, which can be assigned to the well-known concentration quench effect resulted from non-radiative ET between the dopants and lattice instability[38,39]. Detailed QY results are shown in Supplementary Table 2, and the values range from 10.6% to 38.3%.

Figure 2c shows RL spectra of samples with different nominal $Mn^{2+}$ concentrations (0, 3, and 15%). Two very strange phenomena were observed. First, though pure $Cs_3Cu_2I_5$ showed obvious RL, the blue part almost disappeared when only a small amount of $Mn^{2+}$ was doped. There only existed an obvious $Mn^{2+}$ related yellow emission peaked at 564 nm along with a weak blue emission. The small redshift compared to the PL peak is in accord with other halides[19,40]. It is worth noting $Mn^{2+}$ PL is still weak in low doping concentration as shown in Supplementary Fig. 9, but

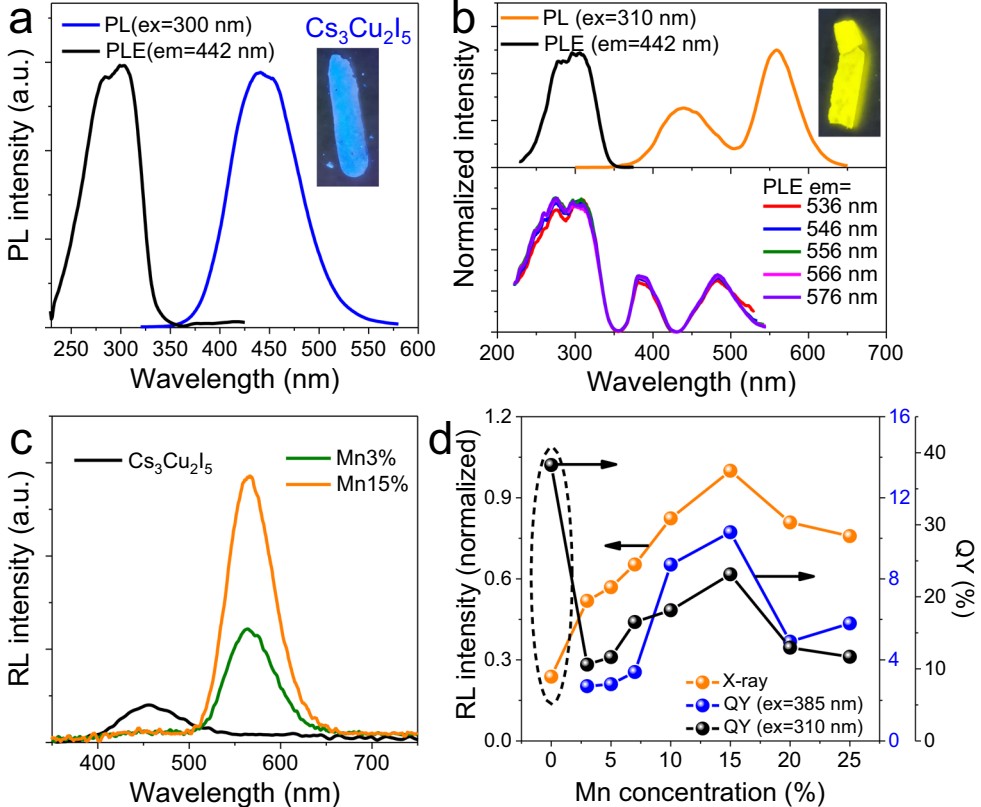

**Fig. 2 Optical and anomalous scintillation properties. a** PL and PLE spectra of undoped Cs₃Cu₂I₅ measured/monitored at 300 and 442 nm. The inset shows the image under 254 nm UV light. **b** PL and PLE spectra of Mn15% measured/monitored at 310/442 nm (top part), and PLE spectra monitored at 536–576 nm (bottom part). The inset shows the image under 365 nm UV light. **c** RL spectra of Cs₃Cu₂I₅:Mn with doping concentrations of 0%, 3%, and 15%. **d** RL intensity and PL QYs of samples under intrinsic (310 nm) and direct Mn²⁺ (385 nm) excitations as a function of doping concentration.

the RL intensity of Mn3% is much higher than that of pure $Cs_3Cu_2I_5$. The RL intensity increases along with the doping concentration and reaches a maximum at a concentration of 15%. Then, a sudden RL intensity reduction emerges when the doping concentration increases furtherly (Supplementary Fig. 10). Second, though pure $Cs_3Cu_2I_5$ possess the highest QY while its RL efficiency is the lowest (Fig. 2d). At low doping concentration (< 7%), the QY under the excitation of 310 nm is relatively low while their RL intensity increases continuously. The variation trend of RL intensity is consistent with that of QY under the excitation of 385 nm (Fig. 2d). Apparently, the RL spectra are completely different from the PL counterparts, implying an interesting luminescence mechanism.

Proposed luminescence mechanism. We then turn our attention to the luminescence mechanism. Though STE emission exhibits strong intensity under the excitation of UV light, it is very weak under X-ray (Fig. 3a). However, when we monitored the luminescence curves under the simultaneous excitation of UV light and X-ray, both the STE and Mn²⁺ emission intensities increased (Fig. 3a). It seems that the X-ray and the UV light can excite the sample independently and the interaction between these two excitation processes can be neglected. Besides, simultaneous excitation did not introduce additional blue emission as its ratio is almost the average of the values under separate excitation (Supplementary Fig. 11a). The emission peak of Mn²⁺ under simultaneous excitation is also the direct combination with a peak in the middle of PL and RL peaks (Supplementary Fig. 11b). In another word, these two excitations are independent to some extent and possess different excitation and recombination mechanisms.

The modulation of reaction parameters contributes to greatly improved RL efficiency (Fig. 3b), and the main difference can be found in the PL/RL curves under different excitations (Supplementary Fig. 12) and the PLE spectra monitored at 556 nm (Supplementary Fig. 13a). Though the difference of PL QY is not very large (Fig. 3b, ex = 310 nm) and the optimized sample does not exhibit the strongest yellow PL (Supplementary Fig. 12a), the samples possessing higher RL efficiencies show apparent enhanced direct Mn²⁺ ($^6A_1(6 S) \rightarrow {}^4T_2(4D)$) excitation efficiency, which accords well with the direct excited PL results (Fig. 3b, ex = 385 nm)[36]. Therefore, the RL of doped samples may be connected with the direct excitation of Mn²⁺ ions. The luminescent properties through the direct excitation based on the $^6A_1(6 S) \rightarrow {}^4T_1(4 G)$ transition (485 nm) show similar behaviors to that of exciting at 385 nm but less significant (Supplementary Fig. 13b).

We make an assumption of the PL and RL mechanism as described in Fig. 3c. It has been reported that the intrinsic blue PL derives from the lattice distortion induced STE (I). Light induced excitation of electron results in the change of $3d^{10}$ to $3d^9$ and the variation of electronic cloud density leads to lattice distortion. This distortion is the origin of STE state and contributes to bright blue emission. After the introduction of Mn²⁺ ions, the lifetime of intrinsic PL decreases along with the increased doping concentration before becoming saturated (Supplementary Fig. 14 and Table 3). The Mn²⁺ related yellow emission possesses two components, indicating two possible exciton ET from STE (II) and CBM (III) levels, as described in Fig. 3c left. In fact, these two recombination processes are in a competitive relation. At low doping concentration, STE related ET dominates and resulted fast

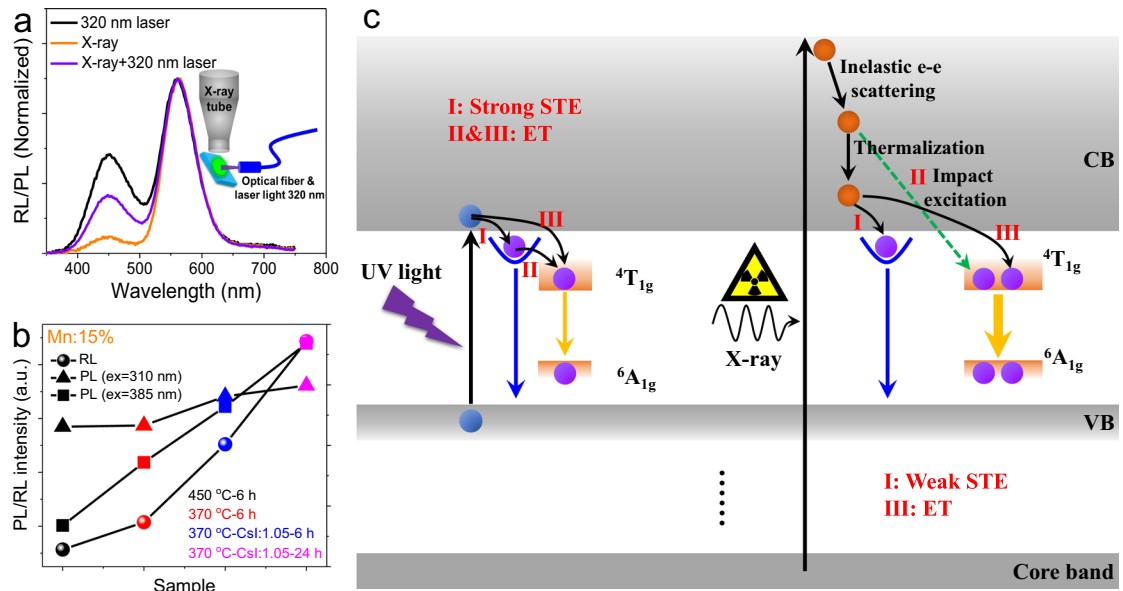

**Fig. 3 Proposed PL and RL mechanism. a** Luminescent spectra of thick Mn15% film under the excitations of X-ray, 320 nm laser or both. The inset shows the measurement system focused on the same point. **b** PL and RL intensity of Mn15% with various reaction parameters under different excitations of X-ray, 310 nm, and 385 nm, respectively. **c** Proposed PL and RL mechanisms including STE, ET, and impact excitation.

decay (Supplementary Fig. 14c and Table 4). With the increase of $Mn^{2+}$ concentration, direct ET from CBM gradually dominates, which possess longer PL decay time.

In contrast to light excitation of band-edge electrons, X-ray excited electrons from core band to much higher energy levels (Fig. 3c right). These electrons then undergo procedures of inelastic electron-electron scattering and thermalization, which finally relax to CBM for recombination. However, since the Cu 3d electrons is not excited directly, lattice distortion is relatively weak compared to that resulted from UV light excitation of Cu 3d electrons. Then, the formation of STE under the excitation of X-ray is very weak, leading to faint blue RL intensity. As a result, majority of the electrons on CBM will transfer to $Mn^{2+}$ levels (III) directly and contribute to strong yellow RL. This might be the main reason for the greatly reduced blue part under the excitation of X-ray compared to the PL curves.

In addition to ET, direct excitation of $Mn^{2+}$ may play a more important role in the RL behavior. First, doped ions can be directly excited through impact excitation (II) provided its concentration is large enough, and when this process occurs, it is strongly dominant. Since the $Mn^{2+}$ doping concentration here is much higher than that of conventional rare earth elements, this mechanism may also make contributions. As we can see, though ET induced yellow emission in 450 °C-6 h sample is the strongest (Supplementary Fig. 12a), its direct excited emission shows the weakest intensity (Fig. 3b). Besides, the RL curves show the same dependence on reaction parameters with the direct excited PL spectra though they possess semblable PL QY under the excitation of 310 nm (Fig. 3b). Namely, the RL mechanism is strongly concerned to the direct excitation of $Mn^{2+}$. However, it is still difficult now to give out the direct evidence of impact excitation and more investigations are needed. The PL QY of samples here are not high enough, implying the existence of enormous non-radiative recombination centers. There is still a large room for future optimization and we believe higher LY values can be achieved with in-depth investigations.

Efficient scintillation properties and robust stability. We conducted detailed investigations to evaluate the RL performance of $Cs_3Cu_2I_5$:Mn powders. $Cs_3Cu_2I_5$:Mn possesses a high X-ray

absorption coefficient comparable to that of CsI, endowing this material with a strong basis for high LY (Supplementary Fig. 15). To accurately measure the RL, equal quality of samples was added in a tailored holder, which was then put into an integrated sphere with a fixed distance to the X-ray source (Fig. 4a). The RL spectra were recorded by a fiber-coupled spectrometer. The LY was measured with a reference strategy and a CsI:Tl single crystal was chosen as the reference. After the optimization of reaction parameters, such as reaction temperature (Supplementary Fig. 16), precursor ratio (Supplementary Fig. 17), and reaction time (Supplementary Fig. 18), the highest RL intensity was achieved at a reaction temperature of 370 °C for 24 h. The optimal reaction temperatures for $Cs_3Cu_2I_5$ and Mn15% are related to their melting points, and the low melting point less than 400 °C can potentially reduce production costs when large area and thick films are fabricated (Supplementary Fig. 19). Figure 4b shows the RL intensity comparison of CsI:Tl single crystal and optimized Mn15%. We can see the Mn15% exhibits a much higher RL intensity with narrower linewidth. Figure 4c indicates the calculated LYs and a reported value of 54,000 ph MeV$^{-1}$ for CsI:Tl single crystal is used as a reference[41]. Despite the LY of $Cs_3Cu_2I_5$ is only 11,900 ph MeV$^{-1}$, the LY of Mn15% is ~67,000 ph MeV$^{-1}$, which is almost the highest value at such a long emission wavelength based on low cost and toxic-element-free precursors. Such a high RL efficiency at long visible wavelength is favorable to NB[22].

Since NBs usually work under extreme conditions, we then assess the thermal stability of the as-prepared samples. Mn15% was chosen as the typical sample for the following studies due to its strongest yellow emission. Figure 4d comparatively shows the temperature-dependent RL intensity of $Cs_3Cu_2I_5$ and Mn15% in a very broad temperature range from 77 to 433 K. We can see that 71% of the RL intensity can be maintained for Mn15% while the undoped one exhibits a significant decrease of 92.5% (Supplementary Fig. 20). The ultra-stability of $Mn^{2+}$ doped scintillators in such a broad temperature variation enables them to adapt to most of the extreme environments, indicating the promising application in outer planets where sudden and severe temperature variation takes place often. In order to demonstrate the sustainable temperature-stability of Mn15%, the integrated RL intensities

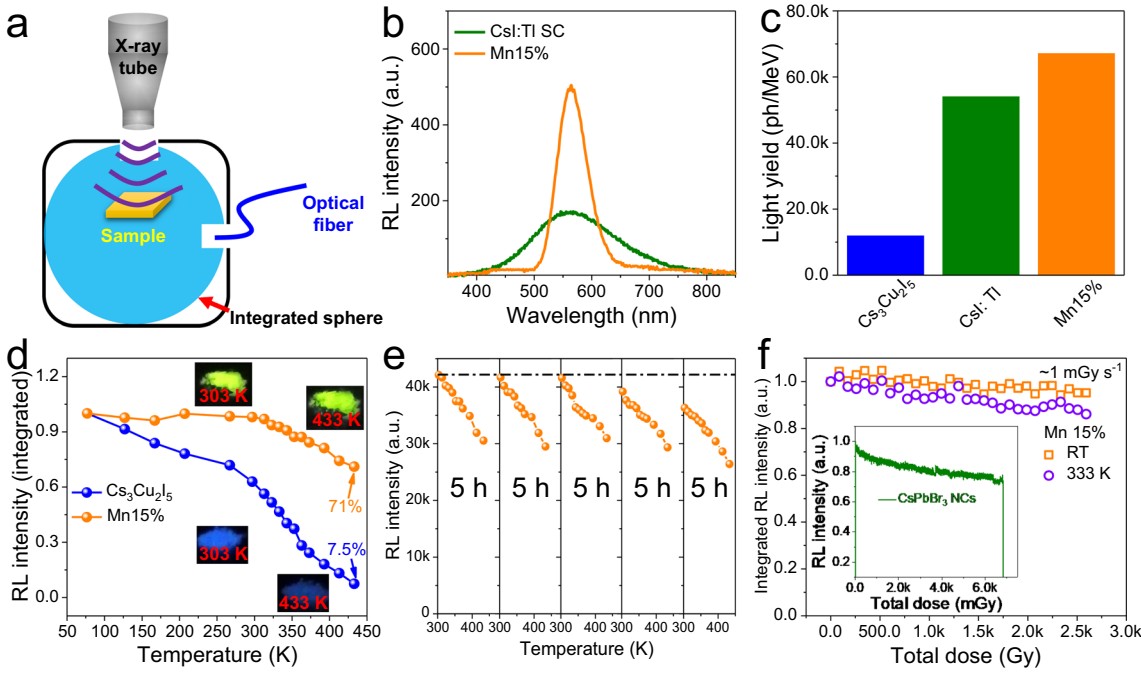

**Fig. 4 Highly efficient and robust scintillation properties. a** Schematic of the light yield measurement system with an integrating sphere. **b** RL spectra of CsI:Tl single crystal and optimized Mn15%. (**c**) Comparison of light yields of CsI:Tl, $Cs_3Cu_2I_5$ and optimized $Cs_3Cu_2I_5$:Mn. **d** RL intensity of $Cs_3Cu_2I_5$ and Mn15% versus temperature at an ultra-large range of 77-433 K. RL stabilities of Mn15% against **e** heating/cooling cycling and **f** long-term operation at room- and high temperature. The inset shows the stability measurement of $CsPbBr_3$ nanocrystals under the same dose rate.

with five heating and cooling cycles in the range from 303 to 433 K were recorded as shown in Fig. 4e without any protection, in which the RL intensity was measured at least 30 min later after the temperature reached the setpoint. For the first three cycles, the intensity remained unchanged. Though further temperature-treating induced RL intensity deterioration resulted from oxidation and phase decomposition, the intensity evolution trends kept the same. The possible deterioration mechanism can be found in Supplementary Fig. 21. All of these results imply the good stability against thermo and ensure good performance of subsequent NBs. The good RL stability is consistent with the PL properties (Supplementary Fig. 22).

Radiation damage is another inevitable issue that all materials have to suffer. To evaluate the radiation stability, we first measured the RL intensity of $Cs_3Cu_2I_5$ under continuous X-ray irradiation (voltage, 50 kV, current, 80 μA, dose rate, ~1 mGy s$^{-1}$). Interestingly, despite the low RL intensity, $Cs_3Cu_2I_5$ showed a persistent scintillation intensity even at 333 K with a total irradiation dose of ~29 Gy (Supplementary Fig. 23), indicating the robust matrix for dopant related emissions. However, as shown in Supplementary Fig. 24, after storage under the ambient condition, the RL exhibited an apparent degradation due to the influence of moisture. The as-prepared powder (Mn 15%) changed from white to light yellow and a phenomenon of agglomeration was observed. When the powder was dispersed in n-butanol, the supernatant became yellow (Supplementary Fig. 25), which was similar to the phenomenon of $I_2$ solids dissolved in n-butanol. It should be noted doping usually cannot modify the chemical property greatly. Therefore, the changed hygroscopy may not result from the doping effect. Considering the hygroscopy of $MnCl_2$ used in the reaction and the existence of unreacted trace-precursor owing to the solid reaction method, we tried purifying the as-prepared powder with n-butanol. Fortunately, $MnCl_2$ exhibits high solubility in n-butanol and this solvent does not destruct the scintillator (Supplementary Figs. 25 and 26). Even after a storage for 2 months without protection, the powder maintained a white

color. No impurity formed and the powder is fine. The crystal structure and RL property almost did not change (Supplementary Fig. 26) after being irradiated and storage. This makes it possible to further measure the stability of this new scintillator.

More than 95% of the initial RL intensity remained for Mn15% after an accumulated irradiation dose of ~2590 Gy at room temperature (Fig. 4f). Since $CsPbBr_3$ scintillator has been extensively studied recently, it was chosen as a reference to determine the stability of Mn15% in the present work. Compared to Mn15%, lead halide perovskite NCs exhibited poor stability under such a high dose rate, which showed a decrease of ~25% just after being irradiated with an accumulated irradiation dose of 6.9 Gy. The stability behaviors of $CsPbBr_3$ NCs observed here are quite different from the early reports, which can be assigned to the high irradiation dose rate used in this work. It is two orders of magnitude higher than that in the previous work[18,19]. Additionally, 86.1% of the initial intensities were reserved when the samples were irradiated by X-ray with the same dose rate at 333 K (total dose of ~2590 Gy). Such robust thermal stability for lead-free halides has never been reported as far as we know. Future investigations should include the protection of these scintillators from the attack of moisture. Fortunately, the sealing technology nowadays is good enough to prevent the invasion of moisture.

**Characterization of nuclear battery.** At this point, we have achieved a highly efficient scintillator with good stability against irradiation and high temperature. Besides, the emission wavelength locates in the region of PVD with high external quantum efficiency, ensuring a good device performance (Fig. 5a)[42]. We then fabricate prototype NBs and the device structure is shown in Fig. 5b. High performance GaAs solar cells were chosen as the PVDs as they not only possess high energy conversion efficiency but also can work under low illuminance conditions. The scintillator layer was fabricated by mixing the powders with polystyrene (0.2 g mL$^{-1}$) and drop-casting. Since the scintillator thickness will either influence the attenuation of X-ray or hinder

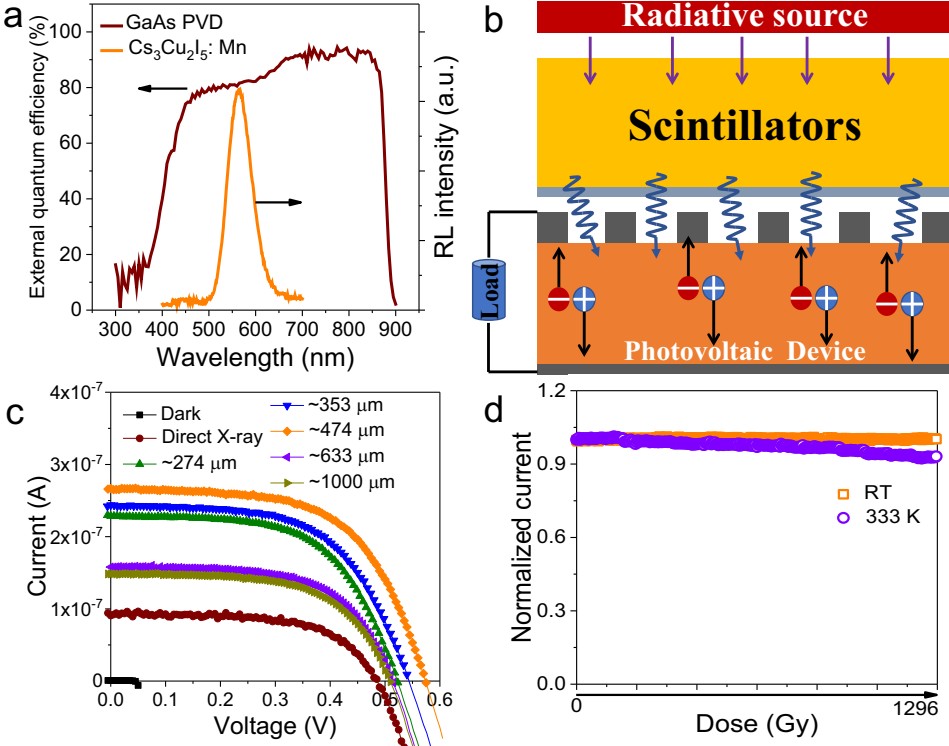

**Fig. 5 High performance NB with robust stability. a** Schematic structure of the radioluminescent NB. **b** Spectral responsivity of a GaAs photovoltaic device and the RL of Mn15% film. **c** *I–V* curves of the NBs with different scintillator thicknesses. **d** Long-term operation stability under room temperature and 333 K.

light output efficiency by scattering, we therefor modulate the film thickness by changing the quantity of Mn15%.

Figure 5c illustrates the current–voltage (*I–V*) results of different NBs with or without Mn15% scintillator layers under the irradiation of X-ray (50 kV, 80 μA). The GaAs PVD sealed with a thin coverslip (0.13 mm) could directly convert X-ray into current with a maximum output power of 0.027 μW. When the $Cs_3Cu_2I_5$: Mn film covered on the device, both the open circuit voltage and short circuit current increased under the same X-ray dose rate. Especially, the short circuit current increased greatly (an increase of 300%), indicating the vast generation of electrons with the assistance of the scintillator layer. Theoretically, the direct type device should exhibit higher efficiency compared to indirect devices due to additional RL step. However, thin film PVDs usually cannot absorb X-ray completely and increasing film thickness is inadvisable due to the limitation of carrier diffusion length and lifetime. Also, it is reported that taking full advantage of the direct and indirect conversion approaches can achieve better device performance[4,40]. Therefore, we should make a balance between scintillator film thickness and X-ray absorption. With the increase of scintillator film thickness, the output power showed a first rising and then decline trend. Obviously, both direct and indirect conversion procedures exist in our device and the optimized output power can reach 0.091 μW, a significant increase of 237% compared to the NB without scintillator. This improvement is much greater than that using $CsPbBr_3$ NCs as the scintillator layer[42]. When other scintillators were applied as the transducer layer, lower output were achieved due to the low LY as shown in Supplementary Fig. 17. We also measured the operation and thermal stabilities of the assembled NBs. Thanks to the high stability of $Cs_3Cu_2I_5$:Mn and the protection of polymer, the NB exhibited a robust power output that after working under ambient conditions with a total irradiation dose of ~1290 Gy (Fig. 5d), no obvious degradation was observed. Besides, 93.1% of the initial current was reserved when the NB worked at 333 K with the same

irradiation dose. Such high stability is seldomly reported for halide-based scintillators.

## Discussion

This work presents a new $Cs_3Cu_2I_5$:Mn scintillator with exceptionally high light yield and robust thermal stability. Mn dopant not only introduces a new emitting center to reduce nonradiative recombination, but also overcomes the temperature quenching effect of intrinsic material. The highly direct excitation efficiency of $Mn^{2+}$, combined with STE and the ET, contribute to a high LY of ~67,000 ph MeV$^{-1}$ at 564 nm, which is the highest value at such a long wavelength based on low-cost elements. Stable matrix and thermal activation of excitons lead to robust thermal stability and irradiation hardness that 71% or more than 95% of the initial RL intensity can be maintained in an ultra-broad temperature range of 77–433 K or after a total irradiation dose of ~2590 Gy, respectively. These superiorities contribute to efficient and stable NBs with an output improvement of 237% respect to that without scintillator layer and a power output of 0.46 μW cm$^{-2}$, which can operate continuously even under high temperature. This work will open a window for the fields of NBs and radiography. Future work should be directed on fabrication of large area film with high quality, scalable fabrication and further performance optimization. Since this material is found to be sensitive to moisture and oxygen (at high temperature), specific packaging technology should be developed for future applications. Additionally, the detailed RL mechanism needs more investigations.

## Methods

**Chemicals**. Cesium iodide (CsI, 99.9%), Cuprous iodide (CuI, 99.5%), and $MnCl_2·4H_2O$ (99%) were purchased from Aladdin. All the chemicals were used without further purification.

**Synthesis of $Cs_3Cu_2I_5$:Mn halides**. A series of $Cs_3Cu_2I_5$:x$Mn^{2+}$ (0.3 ≤ x ≤ 0.25) were synthesized by a solid-state reaction method. The experimental steps were as

follows: CsI, CuI, $MnCl_2.4H_2O$ were put into the mortar or ball grinding tank in a certain proportion to get uniform mixtures. Subsequently, the resulting mixtures were encapsulated in a vacuum quartz tube and placed in a muffle furnace for calcination at 340–450 °C. The reaction time was 6–48 h. After the reaction finished and cooled naturally to room temperature, the calcined product was taken out and grinded to obtain light yellow $Cs_3Cu_2I_5:Mn^{2+}$ powders. Finally, the powder was washed with n-butanol twice to exclude the trace unreacted $MnCl_2$. After centrifugation, the powder was dried under vacuum at 80 degrees for 12 h to obtain the final scintillator material. This purification process is very important otherwise the powders are not stable. Detailed information can be found in the following discussions.

**Fabrication of nuclear batteries**. First of all, a polystyrene (PS) toluene solution ($0.2 \text{ g mL}^{-1}$) was prepared as the matrix. Then, different weights (100–600 mg) of $Cs_3Cu_2I_5:Mn$ 15% were mixed with 1 mL PS solution. The mixtures were stirred rigorously for 4 h and casted on a thin glass substrate for the fabrication of scintillation layers. The thick films were dried under vacuum at room temperature, which were then covered on a GaAs solar cell (1 cm × 1 cm) sealed with a thin glass.

**Characterizations**. Powder X-ray diffraction was characterized by an X-ray diffractometer (Bruker D8 Advance XRD system), and the morphology and elements of $Cs_3Cu_2I_5:Mn$ were detected by a scanning electron microscope (SEM, FEI Quanta 250 F). X-ray photoelectron spectroscopy (XPS) measurements were performed using an achromatic Al Kα source (1486.6 eV) and a double pass cylindrical mirror analyzer (PHI QUANTERA II). The thermal gravimetric analyses were conducted on STA7000, HITACHI. Emission and excitation spectra were collected on Varian Cary Eclipse instrument. The PL lifetimes were measured by FLSP920 (EDINBURGH INSTRUMENTS LTD) equipped with both ns and μs light sources. The absolute quantum yield of the samples was determined using a Quantaurus-QY absolute photoluminescence quantum yield spectrometer (C9920-02G, Hamamatsu Photonics, Japan). The X-ray source is produced from a commercial Amptek mini-x tube with Ag target and a maximum output power of 4W. All the radioluminescence and temperature-dependent spectra were recorded with a fiber spectrometer (QE65PRO, Ocean Optics). The nuclear battery performances were measured with Keithley 2400 under the irradiation of X-ray.

**Computational details**. The structural optimizations and electronic structure calculations are performed based on density functional theory as implemented in VASP code. Exchange correlation energies are considered by the generalized gradient approximation (GGA) using the Perdew-Burke-Ernzerh (PBE) functional. The wave functions are constructed using a projected augmented wave approach with plane wave cutoff energy of 500 eV. The convergence threshold was set as $10^{-4}$ eV in energy and $10^{-3}$ eV/Å in force. The Brillouin zone integration is sampled using a set of 3 × 3 × 3 Monkhorst-Pack k-points.

## Data availability

Source data are provided with this paper. Additional data related to this study are available from the corresponding authors on reasonable request. Source data are provided with this paper.

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

## Acknowledgements

This work was financially supported by NSFC (61874054, 51902160, and 61725402), Young Elite Scientists Sponsorship Program by CAST (2018QNRC001), the Natural Science Foundation of Jiangsu Province (BK20180489, BK20190475), and Fundamental Research Funds for the Central Universities (30918011208).

## Author contributions

X.L. and H.Z. conceived and designed the experiments. X.L., J.C., C.M., and L.J. carried out materials synthesis and characterizations. X.L. constructed the related testing equipment of Figs. 3a and 4a, and drew them. D.Y. and D.G. helped the rietveld refinement. X.C. and Y.W. carried out theoretical simulation. X.L. wrote the paper and all authors discussed the results and commented on the manuscript.

## Competing interests

The authors declare no competing interests.
