## [Peer Review File · Nature Communications]

REVIEWER COMMENTS

Reviewer #1 (Remarks to the Author):

This paper discusses the application of Cs₃Cu₂I₅:Mn scintillator to produce fluorescence in order to enhance their use in nuclear batteries. My comments are given below.

1. In the Abstract, there is a sentence: "...which showed an output improvement of 337% respect to that without scintillator..." it is not easy to decipher from the sentence before, who is compared with whom, and the reasons should be considered.
2. A novel Cs₃Cu₂I₅:Mn scintillator is used to convert the X-ray to radioluminescence. The application of this material is to obtain a nuclear battery with good performance, but the manuscript does not compare the performance of nuclear batteries with other crystalline materials. A comparison of results with other scintillating materials would provide better insights.
3. This type of fluorescent material in this work is not a very new material. How to explain the changes caused by Mn doping in the used Cs₃Cu₂I₅:Mn scintillator? What are the mechanisms and internal reasons for the apparent increase in ultrahigh light yield?
4. The manuscript explores the radiation resistance of the material, using the X-ray generator. The property has dropped about 6% when the radiation dose reaches 38.7 Gy. The dose value is not high enough to prove that it has good irradiation hardness. And the property continues to decline during the irradiation process, and it has not reached a steady trend with the increase of the irradiation dose, so it is not enough to judge the radiation resistance performance. The radiation dose should be increased to analyze related laws.
5. The XRD test result should include the corresponding PDF card, and it may be more conducive to readers to analyze and understand the presence of miscellaneous peaks.
6. The heating/cooling cycling results as shown in Fig. 4e, the decrease of the RL intensity after the first three cycles can't be ignored. And the reason for the degradation is not explained clearly.
7. Page 26, as the author, indicated that the added Mn15% film absorb X-ray and improve the overall luminescence emission intensity. However, in Figure 5c, the relative intensity is decreased after the thickness of the scintillator from 300 mg to 600 mg, please clarify these results. Is there any special meaning to use mg to express material thickness here? What is the appropriate thickness for the effective absorption of rays, and how to determine the optimal thickness?
8. The ordinate data of PL and RL spectra should be reported, such as Supplementary Figures 9-12, 15-17, 19 and 22.
9. Last, in view of semipermanent power source, how to seal the scintillator and photovoltaic device as well as package the fluorescent type nuclear battery that ensures the stability?

Therefore, I don't think the current version of the manuscript meet the high standards for the importance and timeliness of NCOMMS's publication standards.

Reviewer #2 (Remarks to the Author):

Li et. al. reported the smart application of the Cs₃Cu₂I₅:Mn scintillator to fabricate the fluorescent type nuclear battery (NB). Firstly, the Cs₃Cu₂I₅: Mn was reported to exhibit an ultrahigh LY of

~67000 ph/MeV at an emission wavelength of 564 nm. Then, luminescence mechanisms including self-trapped exciton, energy transfer have been proposed. Moreover, Cs₃Cu₂I₅: Mn was studied for the robust thermal stability and irradiation hardness. Finally, the device on the the fabrication of an efficient and stable NB has been demonstrated. As I think, this work is new for the readers in the field of luminescence materials and it will also bring new ideas for the materials scientis working in the field of scintillator and photovoltaic device. I will recommend the publication of this manuscript and some comments are listed below to further improve this manuscript.

1. One of my main concerns is about the discussion of the PL and RL mechanism in Figure 3. Firstly, the authors mentioned that X-ray and the UV light can excite the sample independently, and they possibly posses different excitation and recombination mechanisms. From our related experiment, one can also find the similar emission spectra as X-ray excitation if we change the different excitation wavelength of the UV-visible light. Secondly, considering previous papers on the Mn²⁺ doped metal halides, trap-mediated Mn²⁺ dopant emission was regarded as the mechanism to explain the highly efficient emission (J. Phys. Chem. C 2019, 123, 14239.; J. Phys. Chem. Lett. 2020, 11, 2510). The authors can possibly consider this possible process to explain their findings. Anyway, it is still in the air on the luminescence mechanisms, and the authors are suggested to consider different models.
2. Followed by the first problem, if the trap state accounts for this highly efficient energy transfer and photoluminescence of Mn²⁺. Where is the origin for this trap state? One can find some information from the structure characterization of Mn²⁺ doping. The authors proposed that replacement of Cu ions with Mn ions is possible from both crystal and energy factors. However, how can this system keep the charge balance? It is possible that trap state appeared in such a process. Please describe this model or give your explanation.
3. Energy transfer from STE to Mn²⁺ is not new. Since the decay lifetime for STE is very fast, the lifetime for Mn²⁺ is relatively long. If the energh transfer happens, one should consider this process. Please consider the change of the lifetime in Supplementary Figure 13.
4. In Figure 1f, I don't think the element mapping images can give more evidence on the results.

Reviewer #3 (Remarks to the Author):

Halide perovskite and metal halide materials have recently become potential candidate scintillators, but were mainly applied in the field of X-ray imaging and detection. Li et al.'s work provides a new application for these scintillator materials. They report a novel Mn-doped Cs₃Cu₂I₅ scintillator for the application of nuclear batteries. The authors showed that Mn ions can replace Cu sites by theoretical calculations, then used XRD, EPR, and XPS to experimentally deduce the existence of the Mn dopant. The photoluminescence and radioluminescence properties of samples with different Mn concentration were studied; the best showing a light yield of ~67000 ph/MeV and good thermal stability. As a result, the nuclear battery based on Mn-doped Cs₃Cu₂I₅ scintillator exhibited an output improvement of 337% as compared to that without scintillator.

This paper can be published in Nature Communications once the following questions are addressed:

1. As shown in Fig S2. The diffraction peaks initially shift to higher degrees and then shift to smaller degrees with the increase of Mn concentration. The authors attribute the abnormal lattice contraction results to chloride ions in the precursor However, there is no evidence to prove the existence of Cl ions in the lattice.

In addition, the authors state that "the concentration of chloride ion cannot be too high due to the intrinsic lattice instability". Why not use MnBr₂ instead of MnCl₂ as the Mn source?

2. A series of Cs₃Cu₂I₅ samples with different Mn doping concentrations were prepared. However, the true doping concentration of Mn ions in the lattice was not measured.

Response to the comments on Nature Communications

Manuscript ID: NCOMMS-20-37932A

Dear Reviewers:

We really appreciate the reviewers' valuable comments and suggestions for improving the manuscript. These suggestions not only provide new ideas for the research of this material, but also drive us to think further about the RL mechanism, stability issues and future applications. More necessary characterizations and experiments are added according to your suggestions and the added items are highlighted in red in both of the revised manuscript and supplementary information. Point-by-point responses and changes are listed below:

Reviewer #1: This paper discusses the application of Cs₃Cu₂I₅:Mn scintillator to produce fluorescence in order to enhance their use in nuclear batteries. My comments are given below.

1. In the Abstract, there is a sentence: "...which showed an output improvement of 337% respect to that without scintillator..." it is not easy to decipher from the sentence before, who is compared with whom, and the reasons should be considered.

Our response: We thank the reviewer for the reminding. Here, we want to compare the device covered by a Cs₃Cu₂I₅: Mn scintillator layer with the device of bare GaAs solar cells. Generally, semiconductors can convert high-energy radiation into carriers directly and more efficiently. Indeed, bare GaAs solar cells can also behave as nuclear batteries. However, only very limited rays can be absorbed since bare devices only possess thin active layers (<1 μm) and most of the rays pass through the semiconductor directly, resulting in low conversion efficiency. On the other hand, radioluminescent devices undergo two processes (ray to visible light and light to carriers) and are considered to possess relatively lower energy conversion efficiency, owing to the deficient performance of scintillators. The comparison here can tell the effect and superiority of the scintillator layer, i.e. high-quality scintillators with high light yield can be promising candidates for nuclear batteries.

In the revised manuscript, we change ‘that’ to ‘the photovoltaic device’. Besides, we also add a comparison table in the supplementary information as comment 2 suggested. We will discuss this in the next answer.

2. A novel Cs₃Cu₂I₅:Mn scintillator is used to convert the X-ray to radioluminescence. The application of this material is to obtain a nuclear battery with good performance, but the manuscript does not compare the performance of nuclear batteries with other crystalline materials. A comparison of results with other scintillating materials would provide better insights.

Our response: Thanks very much for the comment. We agree with that comparison of device performance is indeed necessary. In the revised supplementary information, we

add a table (Supplementary Table S5) to compare the effect of the scintillators investigated in this work. Scintillators such as CsPbBr₃ quantum dots, CsPbBr₃ QD/PPO composite, ZnS:Cu and so on, were used in nuclear batteries. We can see that our device shows the highest improvement compared with previous results, confirming the high light yield of Cs₃Cu₂I₅: Mn. It should be noted that we made a mistake in the previous manuscript that the improvement should be 237% rather than 337%. We have modified all the related content in the revised manuscript.

Supplementary Table R1. Device performance comparison of nuclear batteries with different scintillators.

Scintillator	Emission source	Current (μA)	Voltage (kV)	P _{max} without Scintillator	P _{max} (μW)	Increased percentage	Ref.
Cs₃Cu₂I₅:Mn film	X-ray	80	50	0.027	0.091	237%	This work
CsPbBr ₃ Quantum Dot	X-ray	1000	50	1.19	1.34	12.6%	¹
Films							
CsPbBr ₃ QD/PPO solution	X-ray	800	60	0.3	0.8	133%	²
CsPbBr ₃ QD/PPO film	X-ray	800	60	--	1.38×10 ⁻⁶	--	³
ZnS: Cu	X-ray	800	30	0.527	0.797	51.2%	⁴
Ultima Gold	X-ray	500	25	--	0.00478	--	⁵
ZnS: Cu	β-ray	--	--	--	2.1	--	⁶

3. This type of fluorescent material in this work is not a very new material. How to explain the changes caused by Mn doping in the used Cs₃Cu₂I₅:Mn scintillator? What are the mechanisms and internal reasons for the apparent increase in ultrahigh light yield?

Our response: Thanks for this comment, which is related to the luminescence mechanism. To tell the truth, the exact radioluminescence (RL) mechanism is still in the air while we just proposed a possible mechanism according to the luminescence results. It should be noted that though the photoluminescence behavior has been studied before, radioluminescent properties of this material were first reported in this work. The main findings for the RL are the change of the relative intensity of STE and Mn related peaks and apparent increase of light yield after doping. In fact, Mn doping does not induce remarkable PL quantum yield enhancement, indicating the different luminescence mechanisms under the excitation of UV light and X-ray.

Figure R1. (a) PL and PLE spectra of Mn15% measured/monitored at 310 and 556 nm. (b) Demonstration of the measurement and (c) PL and (d) RL spectra from both the front and back sides.

First of all, we make a discussion about the change of the relative emission intensity. As we can see (Figure R1a), there is a small overlap between STE emission and excitation curves. Considering the large penetration depth of X-ray and then deep emission site in the thick scintillator film (hundreds of micrometers), we measured the

PL and RL spectra both from the front and back sides to evaluate the influence of reabsorption or re-excitation (Figure R1b). A decrease of the relative intensity of STE emission can be found when we measure the PL spectrum from the back side. This indicates the re-excitation can influence the output of STE emission from the deep site (Figure R1c). However, the relative RL intensity of STE emission is much weaker than that in PL spectrum no matter what the thickness or the measured side is (Figure R1d), and the change of STE emission intensity can be neglected. This means there is an intrinsic mechanism that plays a more significant effect than reabsorption.

Figure R2. PL spectra and integrated intensities of (a, b) M0%, (c, d) M15% and (e, f) CsPbBr₃ quantum dots.

Reviewer 2 suggested that this may be the results of different excitation wavelengths since X-ray is essentially a beam of light with very short wavelength. Then we measured the PL spectra of M0%, M15% and CsPbBr₃ quantum dots under

different excitation wavelengths. With the decrease of excitation wavelength, both M0 and M15 exhibit a first increase and then decrease trend. On the contrary, CsPbBr₃ quantum dots show a continuous decrease behavior. This may be one of the reasons for the change of relative intensity. However, the excitation and decay processes are totally different under the UV light and X-ray. Explaining this phenomenon from the aspect of UV light excitation is not enough.

In this work, we proposed a possible mechanism to explain the change of relative intensity and the increase of light yield compared to pure Cs₃Cu₂I₅. Since the intrinsic blue PL derives from the lattice distortion induced STE. However, the Cu 3d electrons are not excited directly under X-ray and there are many defect traps in these samples, lattice distortion is relatively weak compared to that resulted from UV light excitation of Cu 3d electrons. Then, the formation probability of STE under the excitation of X-ray is very small, leading to faint blue RL intensity. Besides, the quantum yield of Cs₃Cu₂I₅ prepared here is much lower compared with previous report,⁷ which further induces weak RL. Though Mn doped samples don't have high quantum yield, they still exhibit bright RL. The Mn 3% sample shows a higher light yield compared to Cs₃Cu₂I₅, even it only possesses relatively weak Mn²⁺ related PL. We found that the RL intensity is strongly concerned to the direct excitation of Mn²⁺, and we think the direct excitation of Mn²⁺ ion through impact excitation may be the internal reason. To get more insight into the RL mechanism, RL decay information should be measured while it is unable to get access to this measurement. We are planning to set up specific equipment to investigate the detailed excitation and recombination processes and further investigation is under consideration.

4. The manuscript explores the radiation resistance of the material, using the X-ray generator. The property has dropped about 6% when the radiation dose reaches 38.7 Gy. The dose value is not high enough to prove that it has good irradiation hardness. And the property continues to decline during the irradiation process, and it has not reached a steady trend with the increase of the irradiation dose, so it is not enough to judge the radiation resistance performance. The radiation dose should be increased to

analyze related laws.

Our response: Thanks for this comment. According to the literature and the testing rules in companies, there are two kinds of method to evaluate the radiation resistance of scintillators, irradiating materials with ultrahigh dose rate for a short time or with common dose rate for a long time.⁸ In fact, the destruction ability of high dose rate is not serious due to the low attenuation efficiency. Most of the ray will pass through the materials directly and does not interact with materials. Therefore, manufacturers usually irradiate materials in a low dose rate for a long time. Even though scintillators accept the same dose (1000 Gy, for example), treatment in a low dose rate with long time is more harmful. On the other hand, the powders measured in the manuscript were not sealed. According to the results of supplementary figure 22 and additional results in Figure R5, this material is sensitive to moisture. Therefore, moisture contributes to the fast degradation to some extent, which is also proved by the device stability measurement in Fig. 5d. Encapsulation with polymer improves the radiation resistance greatly.

Figure R3. RL stability against continuous X-ray irradiation with a total dose of ~540 Gy.

To get more insight into the radiation resistance, we encapsulate the powders with polystyrene (PS) in a glovebox and irradiate them for a long time (10 days). The

encapsulated powders still maintain very impressive luminescence intensity (85.8 % of initial value) under continuous irradiation with a total dose of ~540 Gy (Figure R3), indicating its excellent irradiation hardness. It should be noted continuous X-ray irradiation with large dose is really dangerous at university so we have to stop after 10 days. Additionally, the control sample without X-ray irradiation also exhibits a small degradation. Therefore, the deterioration of the scintillation property is the combined effect of X-ray and environmental factors. If the encapsulation technology is good enough, we believe the long-term radiation stability can be further improved, and they can also meet the requirement of practical applications. We also add this results in the revised manuscript.

5. The XRD test result should include the corresponding PDF card, and it may be more conducive to readers to analyze and understand the presence of miscellaneous peaks.

Our response: Thanks for this kind reminding and in the revised manuscript, we add the information of corresponding PDF card as shown in the following figure.

Figure R4. Partial XRD patterns of powders with different doping concentrations and standard diffraction data.

6. The heating/cooling cycling results as shown in Fig. 4e, the decrease of the RL intensity after the first three cycles can't be ignored. And the reason for the degradation is not explained clearly.

Our response: Thanks for your kind reminding. To further understand the deterioration mechanism against heating treatment in the air, we carried out an extreme experiment to treating the sample (Mn15%) at 160 °C for 10 h in the air and in the glovebox (argon), respectively. As shown in Figure R5a, when the powders are treated in the air, they become black after 10 h, indicating the possible decomposition, phase transition, or formation of impurities. On the contrary, the powders treated in the glovebox keep the same after 10 h. This means oxygen or moisture is harmful to this material, especially at such a high temperature. Since water will evaporate at high temperature, oxygen is more destructive to the powders considering the color change.

Figure R5. (a) Pictures of samples treated in the air or in glovebox at 160 °C for 10 h. (b) XRD patterns, (c) Cu 2p, and (d) Mn 2p XPS curves after different treatments.

To understand what happens during the treatment in the air, XRD and XPS measurements are carried out. We can see the XRD pattern of the sample treated in the glovebox almost keeps the same expect for the improved crystallinity (Figure R5b), which is consistent with the picture. However, some diffraction peaks related to impurities (CsI, CuO) emerge for the sample treated in the air, indicating the reaction

of the material with oxygen and decomposition. The oxidation is also proved by the XPS measurement (Figure R5c and d). Therefore, this is the main reason for the degradation after being heated at high temperature under ambient condition. In spite of the material degradation, Diffraction peaks of $\text{Cs}_3\text{Cu}_2\text{I}_5$: Mn still dominate the XRD pattern and the sample still show bright RL under irradiation.. This indicates the oxidation and decomposition happen first on the surface.

On the whole, though this material has shown good thermal stability, the degradation at high temperature under oxygen condition cannot be ignored. This reminds us that excellent encapsulation technology is necessary for practical application and this is also what we should investigate in the future. Comment 9 also mentions this problem and we will discuss more in that answer. For this comment, we add more content in the revised supplementary information file and also make some modification in the main text.

7. Page 26, as the author, indicated that the added Mn15% film absorb X-ray and improve the overall luminescence emission intensity. However, in Figure 5c, the relative intensity is decreased after the thickness of the scintillator from 300 mg to 600 mg, please clarify these results. Is there any special meaning to use mg to express material thickness here? What is the appropriate thickness for the effective absorption of rays, and how to determine the optimal thickness?

Our response: The efficiency of radioluminescent nuclear battery depends on several factors. First of all, the light yield mainly determines the energy conversion efficiency. To absorb adequate X-ray photons, the film should be thick enough. However, since the scintillator films are composed of polycrystalline particles, scattering and possible reabsorption will reduce the light emission at the side of photovoltaic device. Therefore, scintillator films with appropriate thickness are more favorable. Second, it has been evidenced that if the scintillator layer and photovoltaic device absorb X-ray photons and generate photoemission and carriers together, an optimized conversion efficiency can be achieved.⁹ Obviously, the performance of radioluminescent nuclear batteries is not always linear with its thickness.. A thickness that can guarantee high

light emission and ensure enough X-ray photons to pass through is the best. In this work, the film prepared with 600 mg powders exhibit a thickness of $\sim 1000 \mu\text{m}$, which can almost absorb all the X-ray photons generated from the X-ray tube in our lab. Therefore, the light emission is not the best due to strong scattering, and the bottom photovoltaic device does not participate in the direct conversion process. As a result, the conversion efficiency decreases with the increase of film thickness.

In the previous manuscript, we used powder weight to represent our samples just for simplicity as the measured film thickness is not an integer. In the revised manuscript, we have changed them into exact film thickness.

Considering the last question, the optimized film thickness depends on many factors, such as the type of rays and ray photon energy. They exhibit different penetration depth and absorption efficiency. To determine the optimal thickness, we should not only consider above factors, but also take light yield, scattering rate, energy conversion efficiency of the photovoltaic device (from both visible light and rays to carriers), and so on, into consideration. In a previous work, Hong et al. developed an theoretical simulation and related experiment methods to investigate the optimal film thickness considering all above factors.¹⁰ The determination of exact optimal film thickness is out of the scope of this work, we will investigate this in our next research.

8. The ordinate data of PL and RL spectra should be reported, such as Supplementary Figures 9-12, 15-17, 19 and 22.

Our response: Thanks for the kind reminding. In the revised file, we add all the ordinate data of the PL and RL spectra. We also add this information in the main text where it is necessary.

9. Last, in view of semipermanent power source, how to seal the scintillator and photovoltaic device as well as package the fluorescent type nuclear battery that ensures the stability?

Our response: Thanks for this comment, which reminds us to further consider the

factors related to practical applications. As discussed in Comments 4 and 6, and our previous manuscript, this material is sensitive to moisture and oxygen can destroy it at high temperature. It is worth noting that the commonly used GaAs photovoltaic device is very stable under ambient conditions, which has been widely used in space energy conversion. Therefore, for practical applications, we should mainly focus on blocking water and oxygen from the scintillator. Especially, sealing the scintillator films in glovebox will be effective to improve the stability. Besides, the coupling between scintillator and photovoltaic device is important to the performance of final nuclear battery. Packaging them together with well-designed structure will not only lead to improved efficiency, but also enhance the device stability. Fortunately, present encapsulation technology is good enough to inhibit water and oxygen. Both polymer and inorganic materials can be used as binder and packaging glue.¹¹ Additional protect after packaging scintillator and photovoltaic device will further improve the stability. We add one sentence in the conclusion part “Since this material is found to be sensitive to moisture and oxygen (at high temperature), specific packaging technology should be developed for future applications.”.

Therefore, I don't think the current version of the manuscript meet the high standards for the importance and timeliness of NCOMMS's publication standards.

Our response: Thanks very much for your comments and suggestions. They are really helpful and promote us to think more about mechanism, applications and other important issues. Now we have answered all the question and added additional results. We have revised the manuscript accordingly and carefully.

Reviewer #2:

Li et. al. reported the smart application of the Cs₃Cu₂I₅:Mn scintillator to fabricate the fluorescent type nuclear battery (NB). Firstly, the Cs₃Cu₂I₅: Mn was reported to exhibit an ultrahigh LY of ~67000 ph/MeV at an emission wavelength of 564 nm. Then, luminescence mechanisms including self-trapped exciton, energy transfer have been proposed. Moreover, Cs₃Cu₂I₅: Mn was studied for the robust thermal stability and irradiation hardness. Finally, the device on the the fabrication of an efficient and stable NB has been demonstrated. As I think, this work is new for the readers in the field of luminescence materials and it will also bring new ideas for the materials scientis working in the field of scintillator and photovoltaic device. I will recommend the publication of this manuscript and some comments are listed below to further improve this manuscript.

1. One of my main concerns is about the discussion of the PL and RL mechanism in Figure 3. Firstly, the authors mentioned that X-ray and the UV light can excite the sample independently, and they possibly poesses different excitation and recombination mechanisms. From our related experiment, one can also find the similar emission spectra as X-ray excitation if we change the different excitation wavelength of the UV-visible light. Secondly, considering previous papers on the Mn²⁺ doped metal halides, trap-mediated Mn²⁺ dopant emission was regarded as the mechanism to explain the highly efficient emission (J. Phys. Chem. C 2019, 123, 14239.; J. Phys. Chem. Lett. 2020, 11, 2510). The authors can possibly consider this possible process to explain their findings. Anyway, it is still in the air on the luminescence mechanisms, and the authors are suggested to consider different models.

Our response: Thanks for your helpful suggestion that we read the reference papers carefully and reconsider the luminescence mechanism. For the first concern, we measured the PL spectra of M0%, M15% and CsPbBr₃ quantum dots under different excitation wavelengths as shown in **Figure R2**. With the decrease of excitation wavelength, the luminescence intensity of both M0 and M15 increase first and then decrease. On the contrary, CsPbBr₃ quantum dots show a continuous decrease

behavior. As the reviewer suggested, this may be one of the reasons for the change of relative intensity. However, the excitation and decay processes are totally different under UV light and X-ray. Explaining this phenomenon from the aspect of UV light excitation is not enough and further investigations are needed to explain this clearly.

For the second concern, according to previous work and the provided two reference papers, if a trap-mediated energy transfer exists, there are several characteristic features.

First, trap-assisted energy transfer needs specific activation energy, then the Mn^{2+} emission usually exhibits an anti-thermal-quenching behavior that the intensity decreases with the decrease of temperature. Besides, since the trap-assisted transfer and the STE (or the free exciton recombination) emission processes are competitive, they usually show opposite intensity variation trends.¹²⁻¹⁴ However, in this work, both the STE and Mn^{2+} emissions decrease continuously along with the increase of temperature. On the one hand, it seems the trap-mediated process is neglectable. On the other hand, the Mn^{2+} emission may be related to the STE process.¹⁵ Namely, the emission of $\text{Cs}_3\text{Cu}_2\text{I}_5$: Mn comes from the coupling effects of the STE and $^4\text{T}_1 - ^6\text{A}_1$ transition of Mn^{2+} emissions and the Mn^{2+} emission can be induced due to the energy transfers from both free exciton and STE as demonstrated in our previous manuscript. Second, trap-mediated energy transfer can induce significant Mn^{2+} PL enhancement even at a very low doping concentration according to previous work. The STE emission will also decrease after doping. However, for this scintillator, the Mn^{2+} related emission increases slowly with the increase of doping concentration, and the STE emission is still strong at high doping concentration, which are different from previous results. Third, trap-assisted process leads to mono-exponential and almost similar PL lifetime for Mn^{2+} related emission. $\text{Cs}_3\text{Cu}_2\text{I}_5$: Mn shows biexponential decay behavior with both fast and slow recombination routes. It has been demonstrated that the formation of STEs from free excitons occurs on ps time scales, while the transfer time of exciton to Mn^{2+} is around hundreds of ps.¹⁴ Therefore, the fast and slow components may be related to STE- Mn^{2+} and free-exciton- Mn^{2+} emissions, respectively. With the increase of doping concentration, these two

recombination processes dominate one after another. However, as the reviewer said, the luminescence mechanism is still in the air and more detailed investigations are needed to illustrate the exact transition procedures. To get more insight into the RL mechanism, RL decay information should be measured while it is unable to get access to this measurement. We are not setting up specific equipment to investigate the detailed excitation and recombination processes.

2. Followed by the first problem, if the trap state accounts for this highly efficient energy transfer and photoluminescence of Mn^{2+} . Where is the origin for this trap state? One can find some information from the structure characterization of Mn^{2+} doping. The authors proposed that replacement of Cu ions with Mn ions is possible from both crystal and energy factors. However, how can this system keep the charge balance? It is possible that trap state appeared in such a process. Please describe this model or give your explanation.

Figure R6. Excitation-dependent Mn^{2+} and STE emission intensities ($Ex=266$ nm, ambient condition).

Our response: Thanks very much for the above suggestion. First of all, there is no doubt that trap state will form after doping. But, as discussed in the first comment, trap state assisted process can be neglected, or at least does not play the dominating

role in this material. We measured the excitation dependent Mn^{2+} emission intensity to further evaluate the effect of trap state. As shown in Figure R6, both the emission intensity increased with the increase of excitation intensity linearly within a large excitation range, indicating the small effect of trap states. Considering the replacement of Cu^+ with Mn^{2+} , to keep the charge balance, there should be a charge-compensating effect. According to the characterization results, we think there exists copper vacancy. Namely, when a Mn^{2+} ion occupies the site of one Cu^+ , another copper vacancy may form, with the structure disorder of the tetrahedron and the trigon. But, the detailed influence of this phenomenon on the electronic structure and further optical properties cannot be included in this work as there are too many information to be investigated. To further understand the luminescence mechanism, we are now setting up specific equipment to investigate the detailed excitation and recombination processes.

3. Energy transfer from STE to Mn^{2+} is not new. Since the decay lifetime for STE is very fast, the lifetime for Mn^{2+} is relatively long. If the energy transfer happens, one should consider this process. Please consider the change of the lifetime in Supplementary Figure 13.

Our response: Thanks for the suggestion. In the first supplementary information, we made a brief discussion about the change of lifetime. Here, we discuss more about the lifetime according to the reviewer's suggestion and our further consideration. It has been demonstrated that the formation of STEs from free excitons occurs at ps time scales, while the transfer time of exciton to Mn^{2+} is around hundreds of ps. Therefore, the fast and slow components may be related to STE- Mn^{2+} and free-exciton- Mn^{2+} emissions, respectively. First, let us focus on the STE emission. All the samples show a lifetime of $\sim 1 \mu\text{s}$ with small variation. This value is consistent with previous reported results. With the increase of doping concentration, the decay becomes faster, which can be assigned to the fast energy transfer from STE to Mn^{2+} levels.

For the Mn^{2+} emission, at low doping concentration, STE- Mn^{2+} dominates, resulting in short lifetime. With the increase of doping concentration, free-exciton- Mn^{2+}

emission dominates and then leads to longer lifetime. In spite of the dominating role at different doping concentrations, these to emissions exhibit stable decay behavior. Specifically, samples with nominal concentration larger than 15% show similar lifetime and component, indicating the saturation. This is also consistent with other characterizations. Direct excited Mn^{2+} emission possess similar lifetime-scale, this agrees well with previous reports. In the revised manuscript, we add these discussions at the appropriate site.

4. In Figure 1f, I don't think the element mapping images can give more evidence on the results.

Our response: Thanks for your kind suggestion. In this figure, we show the element mapping images to demonstrate that Cs, Cu, I, Mn elements are homogeneously distributed. This indeed cannot tell the doping of Mn^{2+} in the lattice. We move this figure into the supplementary file.

Reviewer #3:

Halide perovskite and metal halide materials have recently become potential candidate scintillators, but were mainly applied in the field of X-ray imaging and detection. Li et al.'s work provides a new application for these scintillator materials. They report a novel Mn-doped Cs₃Cu₂I₅ scintillator for the application of nuclear batteries. The authors showed that Mn ions can replace Cu sites by theoretical calculations, then used XRD, EPR, and XPS to experimentally deduce the existence of the Mn dopant. The photoluminescence and radioluminescence properties of samples with different Mn concentration were studied; the best showing a light yield of ~67000 ph/MeV and good thermal stability. As a result, the nuclear battery based on Mn-doped Cs₃Cu₂I₅ scintillator exhibited an output improvement of 337% as compared to that without scintillator.

This paper can be published in Nature Communications once the following questions are addressed:

Our response: We thank very much for your positive evaluation of our work. The question and suggestions raised by you are helpful. As will be shown below, we have done some more investigations and we believe the quality of the manuscript is improved.

1. As shown in Fig S2. The diffraction peaks initially shift to higher degrees and then shift to smaller degrees with the increase of Mn concentration. The authors attribute the abnormal lattice contraction results to chloride ions in the precursor. However, there is no evidence to prove the existence of Cl ions in the lattice. In addition, the authors state that “the concentration of chloride ion cannot be too high due to the intrinsic lattice instability”. Why not use MnBr₂ instead of MnCl₂ as the Mn source?

Our response: Thanks for your kind comments. First of all, we want to explain why we chose MnCl₂ as our dopant precursor. We mainly consider following two factors. First, as we discussed in the main text, Mn²⁺ (0.66 Å) ion possesses larger ion size than Cu⁺ (0.6 Å). Therefore, to balance the lattice strain resulted from large ion size, anion with smaller size was used instead of I⁻. Besides, MnI₂ is instable, which

decomposes under very low temperature. Second, according to previous reports, $\text{Cs}_3\text{Cu}_2\text{Cl}_5$ usually exhibit much higher quantum yield compared to $\text{Cs}_3\text{Cu}_2\text{Br}_5$ though $\text{Cs}_3\text{Cu}_2\text{Cl}_5$ is not stable at room temperature.^{16, 17} Therefore, we consider that light Cl doping may contribute to the achievement of better luminescence properties. Introducing Br ions into the lattice may lead to reduced luminescence efficiency.

To prove the existence of Cl ions, XPS measurements were carried out as shown in Figure R7a. We can see that the signal to noise ratio of Cl 2p enhanced obviously with the increase of Cl ion content in the sample. Besides, there is no obvious difference for these spectra in shape, indicating the similar Mn-Cl interaction, and then the lattice site of Cl ions in $\text{Cs}_3\text{Cu}_2\text{I}_5$: Mn. The slight shift of the spectrum for MnCl_2 results from the stronger interaction between Mn and Cl ions. Additionally, since the ion size of Cl is much smaller than that of I ion, and only a small amount of Cl doping will contribute to the lattice contraction. This is consistent with the shift of XRD peaks to larger angle and calculation results in Supplementary Figure 3.

We also use MnBr_2 to prepare Mn15% as the reviewer suggested. As shown in Figure R7b, however, the RL intensity of Mn15% doped with MnBr_2 is weaker compared to that of Mn15% doped with MnCl_2 . Interestingly, MnBr_2 doped sample exhibits a small blue shift. To evaluate the positive effect of halide ions, we doping $\text{Cs}_3\text{Cu}_2\text{I}_5$ with other Mn-precursors (MnAc_2 , $\text{Mn}(\text{acac})_2$, MnCO_3 , and MnSO_4) and the RL spectra are shown in Figure R7c. We can see the doping efficiency is very low for MnCO_3 and $\text{Mn}(\text{acac})_2$ and other two precursors result in weak Mn^{2+} related emission. The sample doped with $\text{Mn}(\text{Ac})_2$ even becomes black resulted from the possible carbonization of the organic component.

According to above results and our previous data, Cl ions may contribute to higher RL efficiency indeed. However, as a new material, the exact influence of Cl ions needs more theoretical and experimental investigations, which is beyond the scope of this work. Corresponding results are added in the supplementary file at the appropriate site.

Figure R7. (a) XPS curves of Cl 2p in samples of Mn3%, Mn15%, and pure MnCl₂. (b) Radioluminescence spectra of Mn 15% with doping precursors of MnCl₂ and MnBr₂, respectively. (c) Radioluminescence spectra of Mn 15% with other Mn²⁺ precursors.

2. A series of Cs₃Cu₂I₅ samples with different Mn doping concentrations were prepared. However, the true doping concentration of Mn ions in the lattice was not measured.

Our response: Thanks for your comment. To measure the true content of Mn ions in the sample, we conduct inductively coupled plasma (ICP) measurements and the results are shown in Table R2. The measured results accord well with the nominal concentration except for Mn25% (saturated). It should be noted that since these samples were prepared with a solid-state reaction method without further purification, the measured atom ratio here cannot tell the exact concentration of Mn ions in the

lattice. Some residual precursor may also be included especially for the samples of high doping concentration. According to the XRD patterns, calculated formation energy at high doping concentration, and similar ion radii, we believe the doping concentration can be high enough. To further investigate this interesting material and tell the exact doping concentration, we are now preparing single crystals and nanocrystals with solution methods.

Sample	Nominal atom ratio	Measured atom ratio
Mn3%	3%	3.1%
Mn5%	5%	6.1%
Mn7%	7%	6.9%
Mn10%	10%	10.1%
Mn15%	15%	16.2%
Mn20%	20%	20.7%
Mn25%	25%	36.9%

Reference

1. Xu, Z. et al. CsPbBr₃ Quantum Dot Films with High Luminescence Efficiency and Irradiation Stability for Radioluminescent Nuclear Battery Application. *ACS applied materials & interfaces* **11**, 14191-14199 (2019).
2. Chen, W. et al. Novel radioluminescent nuclear battery: Spectral regulation of perovskite quantum dots. *International Journal of Energy Research* **42**, 2507-2517 (2018).
3. Chen, W. et al. Radioluminescent nuclear battery containing CsPbBr₃ quantum dots: Application of a novel wave-shifting agent. *International Journal of Energy Research* **43**, 4520-4533 (2019).

4. Zhang, Z. et al. Use the Indirect Energy Conversion of the Phosphor Layer to Improve the Performance of Nuclear Batteries. *Energy Technology* **6**, 1959-1965 (2018).
5. Zhang, Z. et al. Application of liquid scintillators as energy conversion materials in nuclear batteries. *Sensors and Actuators A: Physical* **290**, 162-171 (2019).
6. Jiang, T., Xu, Z., Meng, C., Liu, Y. & Tang, X. In-Depth Analysis of the Internal Energy Conversion of Nuclear Batteries and Radiation Degradation of Key Materials. *Energy Technology*, 2000667 (2020).
7. Roccanova, R. et al. Near-Unity Photoluminescence Quantum Yield in Blue-Emitting Cs₃Cu₂Br_{5-x}I_x (0 ≤ x ≤ 5). *ACS Applied Electronic Materials* **1**, 269-274 (2019).
8. Yu, D. et al. Two-dimensional halide perovskite as β-ray scintillator for nuclear radiation monitoring. *Nature Communications* **11**, 3395 (2020).
9. Guo, X. et al. Multi-level radioisotope batteries based on ⁶⁰Co γ source and Radio-voltaic/Radio-photovoltaic dual effects. *Sensors and Actuators A: Physical* **275**, 119-128 (2018).
10. Hong, L., Tang, X.-B., Xu, Z.-H., Liu, Y.-P. & Chen, D. Parameter optimization and experiment verification for a beta radioluminescence nuclear battery. *Journal of Radioanalytical and Nuclear Chemistry* **302**, 701-707 (2014).
11. Tang, X. et al. Physical Parameters of Phosphor Layers and their Effects on the Device Properties of Beta-radioluminescent Nuclear Batteries. *Energy Technology* **3**, 1121-1129 (2015).
12. Pinchetti, V. et al. Trap-Mediated Two-Step Sensitization of Manganese Dopants in Perovskite Nanocrystals. *ACS Energy Letters* **4**, 85-93 (2019).

13. Su, B., Molokeev, M.S. & Xia, Z. Unveiling Mn²⁺ Dopant States in Two-Dimensional Halide Perovskite toward Highly Efficient Photoluminescence. *The Journal of Physical Chemistry Letters* **11**, 2510-2517 (2020).
14. Luo, B. et al. Efficient Trap-Mediated Mn²⁺ Dopant Emission in Two Dimensional Single-Layered Perovskite (CH₃CH₂NH₃)₂PbBr₄. *The Journal of Physical Chemistry C* **123**, 14239-14245 (2019).
15. Zhou, G. et al. Optically Modulated Ultra-Broad-Band Warm White Emission in Mn²⁺-Doped (C₆H₁₈N₂O₂)PbBr₄ Hybrid Metal Halide Phosphor. *Chemistry of Materials* **31**, 5788-5795 (2019).
16. Zhang, R. et al. A Lead-Free All-Inorganic Metal Halide with Near-Unity Green Luminescence. *Laser & Photonics Reviews* **14**, 2000027 (2020).
17. Luo, Z. et al. 0D Cs₃Cu₂X₅ (X = I, Br, and Cl) Nanocrystals: Colloidal Syntheses and Optical Properties. *Small* **16**, 1905226 (2020).

Reviewers' comments:

Reviewer #1 (Remarks to the Author):

In this work the authors demonstrated that the performance of the Cs₃Cu₂I₅ scintillator can be enhanced by adding Mn²⁺. With the doping of manganese, the light yield of the scintillator increases significantly. The high light yield of ~67000 ph MeV⁻¹ at 564 nm was reported. And further use this scintillator in temperature change environment and radiation environment to test its performance changes. These results show a strategy for optimizing the performance of Cs₃Cu₂I₅: Mn scintillator. The subject is of interest. Nevertheless, the treatment is, in my opinion, not appropriate. The main idea is to find a material with high luminous intensity and stable radiation to be used as the transducer component of the battery. However, this kind of light yield and stability are not specifically discussed for the actual application environment of nuclear batteries. What is even more regrettable is that after being exposed to a total irradiation dose of 38.7 Gy, the performance is significantly attenuated. Obviously its radiation stability is worth considering in practical applications. The innovation in luminescent materials is not outstanding, and the energy source of this type of battery is not considered in practical applications. Although authors applied the Cs₃Cu₂I₅:Mn scintillator into the fluorescent type nuclear battery to demonstrate an improvement in their performance, the underlying physics and technologies are already known based on previous studies.

Reviewer #2 (Remarks to the Author):

The authors have properly revised the manuscript, and the paper is suggested to be published in the present form.

Reviewer #1: In this work the authors demonstrated that the performance of the Cs₃Cu₂I₅ scintillator can be enhanced by adding Mn²⁺. With the doping of manganese, the light yield of the scintillator increases significantly. The high light yield of ~67000 ph MeV⁻¹ at 564 nm was reported. And further use this scintillator in temperature change environment and radiation environment to test its performance changes. These results show a strategy for optimizing the performance of Cs₃Cu₂I₅:Mn scintillator. The subject is of interest. Nevertheless, the treatment is, in my opinion, not appropriate. The main idea is to find a material with high luminous intensity and stable radiation to be used as the transducer component of the battery. However, this kind of light yield and stability are not specifically discussed for the actual application environment of nuclear batteries. What is even more regrettable is that after being exposed to a total irradiation dose of 38.7 Gy, the performance is significantly attenuated. Obviously its radiation stability is worth considering in practical applications. The innovation in luminescent materials is not outstanding, and the energy source of this type of battery is not considered in practical applications. Although authors applied the Cs₃Cu₂I₅:Mn scintillator into the fluorescent type nuclear battery to demonstrate an improvement in their performance, the underlying physics and technologies are already known based on previous studies.

Response to the stability issue:

1. Generally, a maintenance of more than 90% is considered to be a good stability. As shown in our initial manuscript, 94% and 93% of the initial intensities were reserved when the samples were irradiated by X-ray with the same dose rate at 333 K (total dose of 38.7 Gy) and 373 K (total dose of 26.4 Gy), respectively. Besides, these results were measured at high temperature without any protection. For the measurement of stability, we have explained in our first response letter. According to the testing rules in companies (iRay Technology, Shanghai), there are two kinds of method to evaluate the radiation resistance of scintillators, irradiating materials with ultrahigh dose rate, such as γ -ray, for a short time or with common dose rate for a long time. Measurement in a low dose rate with long time is more harmful to scintillators.
2. The measurement of 38.7 Gy was our initial result. We have added the results of encapsulated sample in the first response. After being irradiated with a total dose of ~540 Gy, an intensity of 85.8% compared with its initial value can be achieved. This performance is better than most of recently reported work and obviously, the stability

can be further improved with more investigations.

3. Most importantly, in the past several months, we were continuously working on the improvement of stability and we figured out the deterioration mechanism and tried stabilizing the scintillators successfully.

It has been reported that $\text{Cs}_3\text{Cu}_2\text{I}_5$ possess excellent stability under various conditions. Our results also prove this as shown in the SI file. However, as shown in **Figures R1a and b**, if the as-prepared powder (Mn 15%) was stored without any treatment, the color changed from white to light yellow and a phenomenon of agglomeration was observed. When the powder was dispersed in n-butanol, the supernatant became yellow (**Figure R1c**), which was similar to the phenomenon of I_2 solids dissolved in n-butanol (**Figure R1d**). On the other hand, as discussed in the 1st round of review, the stability can be improved significantly with the protection of glovebox or polymer. Besides, the luminescence efficiency can be recovered after being dried under vacuum at high temperature.

According to above observations, we considered that moisture contributed greatly to the deterioration. It should be noted doping usually cannot modify the chemical property greatly. Therefore, the changed hygroscopy may not result from the doping effect. Considering the hygroscopy of MnCl_2 used in the reaction and the existence of unreacted trace-precursor owing to the solid reaction method, we tried purifying the as-prepared powder with n-butanol. Fortunately, MnCl_2 exhibits high solubility in n-butanol and this solvent doesn't dissolve or destruct the scintillator as evidenced in **Figure R1e and Figure R2a**. Even after a storage for two months without protection, the powder maintained a white color without agglomeration (**Figures R1f**). No impurity formed as demonstrated in **Figures R1g** and the powder is fine. The crystal structure and RL property almost did not change as evidenced in **Figure R2** after being irradiated and storage. This makes it possible to further measure the stability of this new scintillator.

We then prolong the irradiation time to check the stability of the new scintillator and the related battery as shown in **Figure R3**. Interestingly, after a total dose of ~2590 Gy with about one month irradiation, 95.2% of the RL intensity was maintained and 86.1% of the initial intensity was reserved even at a high temperature (**Figure R3**, 333 K). Accordingly, the current almost did not change after an irradiation time of half a month at room temperature and 93.1% of the current was maintained at 333 K under the same dose (~1290 Gy). Obviously, the stability of the scintillator was improved greatly after purification and no impurity formed as shown in **Figures R1h and i**. We revised the manuscript according to above results and also added some descriptions in the experimental section.

Such a good stability, in our opinion, is better than most of recent work. We hope these additional results can satisfy the reviewer.

Figure R1. Illustration of the deterioration phenomena (a-d), improved stability after purification (e-g), and improved irradiation hardness of the sample (h and i).

Figure R2. (a) XRD patterns of as-prepared, washed and irradiated Mn 15% powder. (b) RL spectra of corresponding samples.

Figure R3. (a) RL stability of the purified sample with a high dose at room temperature and 333 K. (b) Long-term operation stability of the battery under room temperature and 333 K.

Response to “Nevertheless, the treatment is, However, this kind of light yield and stability are not specifically discussed for the actual application environment of nuclear batteries.”

We think this is similar to the sentence in reason #3 (the energy source of this type of battery is not considered in practical applications).

Though the reviewer did not explain clearly, we think the reviewer was worrying about the radioactive sources. It is true that the easily achieved sources are β - and γ -ray sources. In this work, we use X-ray instead of γ -ray due to the easy acquirement of X-ray in university labs. Besides, X- and γ -rays are similar except their photon energy. Demonstrating the performances of the scintillator and device with X-ray can be directly referenced by the application condition with γ -ray. Furthermore, the main point of this work is the demonstration of a new scintillator with high light yield and good stability. We think it is reasonable to demonstrate the advantages of the material like this. Regarding to the measurement of stability, we believe it is also reasonable since it was under the guidance of a professional company as discussed in response 1 and our first response letter.

Response to “The innovation ... outstanding, ... the underlying physics and technologies are already known based on previous studies.”

1. This material, to the best of our knowledge, is used as a scintillator for the first time. Additionally, it exhibits high light yield and good stability, which are better than conventional scintillators. The light yield is the highest value at such a long emission wavelength based on low-cost, rare earth or toxic element free materials (<http://scintillator.lbl.gov/>). What is more, this scintillator improves the device performance significantly as the reviewer said.

2. The scintillator here exhibits an abnormal luminescence behavior, and a possible radioluminescence mechanism was proposed. This is one of the new scientific things of this work. Importantly, the proposed luminescence mechanism may give a reference for future scintillator design and development. For example, based on this work, we developed some other new scintillators which show high light yield, fast luminescence decay, and good stability (unpublished results). They exhibit similar luminescence phenomenon and can be explained by this mechanism.

Figure R4. Other newly developed halide scintillators of other (a) doping and (b) material systems.

3. Besides, we want to point out that though the working principle and underlying physical and technological things of solar cells, lithium batteries, light-emitting devices, X-ray related equipment, etc. have been investigated for decades and most of them are already known. They are still under intense investigations. The main point of this work is to develop new materials and improve device performance and we indeed achieve this. We think this can also be called as innovation. To further emphasize this point, we give out a reference. Though energy transfer is an old concept which has been studied in enormous material systems and applications. However, when it was used in the common CsPbBr_3 perovskite nanocrystal scintillator and improve the performance, it was still interesting (Nat. Nanotech. 2020, 15, 462-468). We not only reported a new scintillator here, but also proposed a mechanism for the abnormal phenomenon and the development of new scintillators.

Reviewer #2: The authors have properly revised the manuscript, and the paper is suggested to be published in the present form.

Response: Thank you very much for your positive comments.

REVIEWER COMMENTS

Reviewer #2 (Remarks to the Author):

This is a revised or re-submitted manuscript. I have previously read this manuscript, as well as the new response and revision. I think the quality of this manuscript is improved and it can be considered for the publication in NC.

Reviewer #3 (Remarks to the Author):

The authors have done additional experimental work to respond to the reviewer. This manuscript can potentially be accepted after a minor revision that addresses the following comments:

1) The photoluminescence excitation spectrum in Fig 2b should be measured for both the intrinsic emission and the Mn²⁺ emission at wavelength positions where the two peaks don't overlap. The authors should present an interpretation of the results, and whether it supports their hypothesis of a single phase or not.

2)As a control, the authors should experimentally compare the performance of the NB using Cs₃Cu₂I₅:Mn in Figure 5 to an NB using a state-of-the-art scintillator in the same configuration.

Reviewer #2: This is a revised or re-submitted manuscript. I have previously read this manuscript, as well as the new response and revision. I think the quality of this manuscript is improved and it can be considered for the publication in NC.

Response: Thank you very much for your positive comments.

Reviewer #3: The authors have done additional experimental work to respond to the reviewer. This manuscript can potentially be accepted after a minor revision that addresses the following comments:

Response: Thank you very much for your positive comments.

1) The photoluminescence excitation spectrum in Fig 2b should be measured for both the intrinsic emission and the Mn²⁺ emission at wavelength positions where the two peaks don't overlap. The authors should present an interpretation of the results, and whether it supports their hypothesis of a single phase or not.

Response: Thanks for your suggestion. First, on the one hand, according to the XRD patterns and related analyses, the scintillator is a single phase. On the other hand, the high resolved sextet spectral feature suggests a uniform dispersion of Mn²⁺ in lattice sites rather than the formation of other Mn²⁺ related phases. Second, generally, for Mn²⁺ based metal iodides, the optimized excitations are transitions of ${}^6A_1(6S) \rightarrow {}^4T_2(4D)$ and ${}^6A_1(6S) \rightarrow {}^4T_1(4G)$.¹⁻³ However, the best excitation of Cs₃Cu₂I₅:Mn is ~310 nm here, which is the intrinsic PLE peak of Cs₃Cu₂I₅.

Third, as the reviewer suggested, we then show photoluminescence excitation spectra (PLE) of both the intrinsic emission and the Mn²⁺ emission at different wavelengths. As shown in Figure R1 and revised Figure 2b, for Mn15%, when we measured the PLE spectrum with a monitor wavelength of 442 nm (intrinsic emission), it exhibited a similar spectrum to that of undoped sample (Figure 2a). When we measured the PLE spectra with monitor wavelengths of 536-576 nm, they exhibited similar shape as well as some other features. The UV part shorter than 350 nm and that monitored at intrinsic emission wavelength are semblable, indicating the efficient excitation of Mn²⁺ related emission from intrinsic energy band. This implies the energy transfer from STE to Mn²⁺ related energy levels. Additionally, the similar shape of PLE spectra with different monitor wavelength indicates the single excited state. If there are several states, the PLE spectra may show a varied structure. On the whole, the samples investigated here possess a single phase.

Figure R1. PL and PLE spectra of Mn15% measured/monitored at 310/442 nm (top part), and PLE spectra monitored at 536-576 nm (bottom part). The inset shows the image under 365 nm UV light.

2)As a control, the authors should experimentally compare the performance of the NB using $\text{Cs}_3\text{Cu}_2\text{I}_5:\text{Mn}$ in Figure 5 to an NB using a state-of-the-art scintillator in the same configuration.

Response: Thanks for your suggestion. According to previous reports, we add the device performance applied with ZnS: Cu, CsI: Tl, and CsPbBr_3 QDs as shown in Figure R2. We also add the results in the supplementary file. We can see that CsPbBr_3 QDs shows an output improvement of 26.2%, which is comparable with previous report. In fact, CsPbBr_3 QDs possess low light yield due to the severe self-absorption effect. ZnS: Cu and CsI: Tl are reported to possess higher light yield, which endow the devices with higher output performance, and improvement of 106% and 214% are achieved, respectively. These results are consistent with the high light yield and long emission wavelength of $\text{Cs}_3\text{Cu}_2\text{I}_5:\text{Mn}$.

Figure R2. I-V curves of NBs with different scintillators.

Reference

1. Jiang, T. et al. Highly Efficient and Tunable Emission of Lead-Free Manganese Halides toward White Light-Emitting Diode and X-Ray Scintillation Applications. *Advanced Functional Materials* **31**, 2009973 (2021).
2. Morad, V. et al. Manganese(II) in Tetrahedral Halide Environment: Factors Governing Bright Green Luminescence. *Chemistry of Materials* **31**, 10161-10169 (2019).
3. Su, B., Molokeev, M.S. & Xia, Z. Mn²⁺-Based narrow-band green-emitting Cs₃MnBr₅ phosphor and the performance optimization by Zn²⁺ alloying. *Journal of Materials Chemistry C* **7**, 11220-11226 (2019).

REVIEWERS' COMMENTS

Reviewer #3 (Remarks to the Author):

The authors have addressed all my comments. I recommend the publication of this manuscript.

Response to the comments on Nature Communications

Manuscript ID: NCOMMS-20-37932C

Dear editor and reviewers:

We sincerely thank you for your time. Your comments enhanced this work indeed, and help us understand this new material in-depth.

Reviewer #3: The authors have addressed all my comments. I recommend the publication of this manuscript.

Response: Thank you very much for your positive comments.